# mTORC1 Signaling in AgRP Neurons Is Not Required to Induce Major Neuroendocrine Adaptations to Food Restriction

**DOI:** 10.3390/cells12202442

**Published:** 2023-10-12

**Authors:** Gabriel O. de Souza, Pryscila D. S. Teixeira, Niels O. S. Câmara, Jose Donato

**Affiliations:** 1Departamento de Fisiologia e Biofisica, Instituto de Ciencias Biomedicas, Universidade de Sao Paulo, Sao Paulo 05508-000, SP, Brazil; gabrielorefice@usp.br (G.O.d.S.); teixeirapds@gmail.com (P.D.S.T.); 2Departamento de Imunologia, Instituto de Ciencias Biomedicas, Universidade de Sao Paulo, Sao Paulo 05508-000, SP, Brazil; niels@icb.usp.br

**Keywords:** agouti-related protein, food intake, hypothalamus, mammalian target of rapamycin, mouse, raptor, sex differences

## Abstract

Hypothalamic mTORC1 signaling is involved in nutrient sensing. Neurons that express the agouti-related protein (AgRP) are activated by food restriction and integrate interoceptive and exteroceptive signals to control food intake, energy expenditure, and other metabolic responses. To determine whether mTORC1 signaling in AgRP neurons is necessary for regulating energy and glucose homeostasis, especially in situations of negative energy balance, mice carrying ablation of the *Raptor* gene exclusively in AgRP-expressing cells were generated. AgRP^ΔRaptor^ mice showed no differences in body weight, fat mass, food intake, or energy expenditure; however, a slight improvement in glucose homeostasis was observed compared to the control group. When subjected to 5 days of food restriction (40% basal intake), AgRP^ΔRaptor^ female mice lost less lean body mass and showed a blunted reduction in energy expenditure, whereas AgRP^ΔRaptor^ male mice maintained a higher energy expenditure compared to control mice during the food restriction and 5 days of refeeding period. AgRP^ΔRaptor^ female mice did not exhibit the food restriction-induced increase in serum corticosterone levels. Finally, although hypothalamic fasting- or refeeding-induced Fos expression showed no differences between the groups, AgRP^ΔRaptor^ mice displayed increased hyperphagia during refeeding. Thus, some metabolic and neuroendocrine responses to food restriction are disturbed in AgRP^ΔRaptor^ mice.

## 1. Introduction

Several hypothalamic neuronal populations regulate energy and glucose homeostasis [1]. However, neurons that express the agouti-related protein (AgRP) play a critical role in this aspect. AgRP neurons co-express the neuropeptide Y (NPY) and are exclusively found in the ventromedial arcuate nucleus of the hypothalamus (ARH), which is near the median eminence, where the blood-brain barrier is less selective. Consequently, AgRP neurons are susceptible to variations in the systemic bloodstream’s nutrients, cytokines, and hormones. Thus, AgRP neurons are specialized in integrating interoceptive and exteroceptive signals to control food intake, energy expenditure, and other metabolic aspects [1]. The activation of AgRP neurons drives food intake, even in previously satiated mice, and reduces energy expenditure [2]. Therefore, AgRP neurons are considered an essential neuronal population that promotes hunger and activates energy conservation processes, especially in situations of negative energy balance (e.g., fasting or chronic food deprivation).

Several intracellular signaling pathways are recruited in AgRP neurons by different hormonal inputs or changes in nutrient levels. Activating these signaling pathways controls the firing activity, gene expression, and other cellular aspects of AgRP neurons. For example, leptin recruits the signal transducer and activator of the transcription-3 (STAT3) pathway [1,3,4,5]. STAT3 ablation in AgRP neurons leads to obesity, hyperleptinemia, mild hyperphagia, and reduced responsiveness to leptin [3]. On the other hand, the MAPK MEK/ERK pathway mediates the action of insulin to repress *Npy*/*Agrp* gene expression [6]. The effects of insulin and leptin on AgRP neurons are also integrated by the phosphoinositide 3-kinase (PI3K)/Akt signaling pathway [7,8]. Ablation of the p110β PI3K subunit in AgRP neurons reduces body weight, food intake, fat mass, and leptinemia in mice [9]. Another study has shown that genetic inactivation of the STAT5 pathway in AgRP neurons produces mild obesity in female mice but not in males [10]. Furthermore, several neuroendocrine responses to food restriction, including the upregulation of the hypothalamic *Agrp* gene expression or serum corticosterone levels, were attenuated in mice carrying ablation of the STAT5 pathway in AgRP neurons [10]. Thus, AgRP neurons rely on STAT3, STAT5, PI3K, and MAPK signaling pathways to regulate energy homeostasis via interoceptive cues.

The mammalian target of rapamycin complex 1 (mTORC1) is a protein complex responsible for sensing nutrients and consequently controlling protein synthesis and other metabolic aspects [11]. Raptor is an essential protein associated with mTORC1 [12]; therefore, the inactivation of Raptor causes a loss of function of mTORC1 signaling [13]. Several studies have demonstrated that the hypothalamic mTORC1 pathway regulates food intake and body adiposity [14,15,16,17]. AgRP neurons are involved in the effects of mTORC1 on energy homeostasis. Amino acid availability, particularly leucine, recruits mTORC1 signaling and regulates *Agrp* gene expression and food intake in vivo [18]. The hormone ghrelin activates the mTORC1 signaling pathway in the ARH [19]. Central inhibition of mTORC1 blunts the orexigenic action of ghrelin, which depends on AgRP neurons [20,21], and prevents ghrelin’s effects on *Agrp* and *Npy* gene expression [19]. Raptor ablation in AgRP neurons does not affect energy homeostasis in ad libitum-fed mice but disturbs the circadian gene expression of *Agrp* and *Npy* [22]. Another study demonstrated that the mTORC1 signaling pathway in AgRP neurons regulates interscapular brown adipose tissue (BAT) thermogenesis and energy expenditure [23].

Despite the robust evidence that mTORC1 signaling in AgRP neurons is associated with the control of energy homeostasis, it is currently unclear whether this signaling pathway is necessary to produce significant neuroendocrine responses to situations of negative energy balance. It is worth mentioning that fasting/food restriction induces the activation of AgRP neurons [4,24,25,26]; therefore, the primary physiological effects of AgRP neurons are usually associated with situations of negative energy balance. Thus, the objective of the current study was to investigate whether mTORC1 signaling in AgRP neurons is necessary for regulating energy and glucose homeostasis, particularly in situations of negative energy balance. For this purpose, mice carrying ablation of the *Raptor* gene exclusively in AgRP-expressing cells were generated, and possible alterations in metabolism were determined in situations of ad libitum food intake, during food restriction, and in a refeeding period. Considering the well-known sexual dimorphism exhibited by ARH neurons [27] and the sex differences in the responses to different metabolic challenges [24,28,29,30], the current study evaluated both male and female mice.

## 2. Materials and Methods

### 2.1. Mice

To generate mice carrying disruption of mTORC1 signaling specifically in AgRP neurons, AgRP-Cre mice (The Jackson Laboratory, Bar Harbor, ME, USA; Stock No. 012899) were initially crossed with animals carrying a pair of loxP sites flanking the *Raptor* gene (The Jackson Laboratory; Stock No. 013188). After successive breedings, Raptor^flox/flox^ mice carrying the AgRP-Cre gene were generated (AgRP^ΔRaptor^ mice), whereas negative littermates for the Cre gene were considered control mice. AgRP-reporter mice were generated as previously described [31]. Both males and females were used in the experiments. Mice were weaned and genotyped within 3–4 weeks of life. The mutations were confirmed by PCR using the REDExtract-N-Amp™ Tissue PCR Kit (Sigma-Aldrich, St. Louis, MO, USA). The animal room was maintained on a 12 h light/dark cycle. A regular rodent diet (2.99 kcal/g; 9.4% kcal derived from fat; Nuvilab CR-1, Quimtia, Brazil) was provided to the mice. The experimental procedures were approved by the Ethics Committee on the Use of Animals of the Instituto de Ciencias Biomedicas at the Universidade de Sao Paulo (protocol 73/2017, approved on 7 July 2017).

### 2.2. Immunofluorescence

To demonstrate the absence of mTORC1 signaling in ARH^AgRP^ neurons, 16-hour (overnight) fasted and refed mice (2 h of refeeding after overnight fasting) were perfused, and their brains were processed, as previously described [24,31]. AgRP-reporter mice (*n* = 3) were also perfused after 2 h of refeeding to demonstrate the expression of the phosphorylated form of S6 (pS6) in AgRP neurons. Then, brain sections were rinsed in 0.02 M potassium phosphate-buffered saline, pH 7.4 (KPBS), and blocked for 1 h in 3% normal serum. Sections were incubated in an anti-pS6 antibody (1:1000; Cell Signaling Technologies, Danvers, EUA; Cat# 5364) overnight. Alexa Fluor^488^-conjugated secondary antibody (1:500; Jackson ImmunoResearch Laboratories, West Grove, PA, USA) was used to label pS6-positive cells. The brains were mounted onto gelatin-coated slides, which were covered with Fluoromount G (Electron Microscopic Sciences, Hatfield, PA, USA). The same brain series was used to investigate the expression of Fos, a marker of neuronal activation, in different hypothalamic nuclei of fasted and refed mice. Fos was labeled using the anti-Fos antibody (1:20,000; Millipore; Cat# Ab5; RRID: AB_2314043). Photomicrographs were obtained using an Axioimager A1 microscope (Zeiss, Munich, Germany) connected with a Zeiss Axiocam 512 camera. The number of pS6-positive cells was determined in the ARH of control and AgRP^ΔRaptor^ mice, either in fasted or refed animals. The number of Fos-positive cells was determined in several hypothalamic nuclei.

### 2.3. Metabolic Measurements

Body weight and body composition were analyzed every week from weaning until 16 weeks of life. Body composition was analyzed by TD-NMR (LF50 body composition mouse analyzer, Bruker, Germany). For the glucose and insulin tolerance tests, food was removed from the cage 4 h before each test. Mice received 2 g glucose/kg or 1 IU insulin/kg for the glucose tolerance test (GTT) and insulin tolerance test (ITT), respectively. A glucose meter was used to determine blood glucose levels during these tests using blood samples from the tail.

### 2.4. Food Restriction Protocol

Before food restriction, mice were single-housed, and their average food intake was determined. To induce food restriction, mice received 40% of their basal food intake for 5 consecutive days, 2 h before lights went off, followed by 5 days of ad libitum refeeding. Food intake, body weight, body composition, and glycemia were determined at baseline and then daily during food restriction (when food was provided) and in the refeeding period. The Oxymax/Comprehensive Lab Animal Monitoring System (Columbus Instruments, Columbus, OH, USA) was used to determine O_2_ and CO_2_ levels and ambulatory activity by infrared sensors. The respiratory exchange ratio (RER; CO_2_ production/O_2_ consumption) was calculated.

### 2.5. Tissue Analysis

The entire hypothalamus was collected in a group of ad libitum-fed mice and in mice after 2 days of food restriction. Total RNA extraction, complementary DNA synthesis, and quantitative real-time PCR were performed following a previously described protocol [24,26,31,32]. The following primers were used: *Actb* (forward: gctccggcatgtgcaaag; reverse: catcacaccctggtgccta), *Agrp* (forward: ctttggcggaggtgctagat; reverse: aggactcgtgcagccttacac), *Pomc* (forward: atagacgtgtggagctggtgc; reverse: gcaagccagcaggttgct), and *Ppia* (forward: ccgttcagctctgggatgac; reverse: gggcagcccagaacatcat). The geometric average of *Actb* and *Ppia* expressions was used to normalize the samples. mRNA levels were calculated by 2^−ΔΔCt^. Enzyme-linked immunosorbent assays were used to determine the serum concentrations of T4 (Calbiotech, El Cajon, USA, # T4044T-100) and corticosterone (Arbor Assays, Ann Arbor, USA, #K014- H1). T4 and corticosterone kits have a limit of detection determined as 25 µg/dL and 16.9 pg/mL and an intra- and inter-assay coefficient of variability ≤8% and ≤6%, respectively.

### 2.6. Statistical Analysis

The unpaired two-tailed Student’s *t*-test or two-way ANOVA plus the Newman–Keuls multiple comparisons test determined differences between the experimental groups, depending on the number of variables analyzed. Changes over time were determined by repeated measures, two-way ANOVA, and Sidak’s multiple comparisons test. The Prism software (GraphPad, San Diego, CA, USA) was used for the statistical analyses. The results are expressed as the mean ± standard error of the mean. The statistical tests and sample sizes are found in each figure legend.

## 3. Results

### 3.1. Generation of Mice Carrying Ablation of the mTORC1 Signaling in ARH^AgRP^ Neurons

mTORC1 signaling is involved in nutrient sensing [11,14]. So, changes in nutritional status regulate the phosphorylation of proteins in the mTORC1 pathway. S6 protein is a major downstream effector activated by mTORC1 signaling [15,16]. Thus, to demonstrate the absence of mTORC1 signaling in ARH^AgRP^ neurons, we subjected control and AgRP^ΔRaptor^ mice to 16-hour fasting or 2-hour refeeding and analyzed the number of ARH cells expressing S6 phosphorylation (pS6-positive cells). In accordance with previous studies [22], a small number of pS6-positive cells were found in the ARH of fasted mice (Figure 1A,B). In control mice, refeeding induced a robust increase in pS6-positive cells in the ventromedial ARH, where AgRP neurons are found (Figure 1C). In contrast, AgRP^ΔRaptor^ mice could not respond to refeeding, maintaining a small number of pS6-positive cells in the ARH, similar to fasted mice (Figure 1D,E). Since pS6-positive cells were not colocalized with AgRP neurons, a group of AgRP-reporter mice (*n* = 3) was also subjected to refeeding, and the coexpression between pS6 and AgRP (indirectly visualized by a reporter protein) was determined. We observed that 90.2% ± 0.9% of pS6-positive cells in the ARH were AgRP neurons (Figure 1F–H). Thus, variations in nutritional status affect mTORC1 signaling predominantly in ARH^AgRP^ neurons.

### 3.2. AgRP^ΔRaptor^ Mice Show Normal Body Weight but a Slight Improvement in Glucose Homeostasis

Hypothalamic mTORC1 signaling regulates several metabolic aspects in vivo [14,15,16,17,18,19,22,23,33,34,35]. Thus, we investigated the metabolic phenotype of AgRP^ΔRaptor^ mice. Male and female AgRP^ΔRaptor^ mice exhibited no changes in body weight, fat mass, or lean body mass compared to control mice (Figure 2). Then, glucose homeostasis was evaluated by a GTT and an ITT. Male AgRP^ΔRaptor^ mice show no difference in the GTT (Figure 3A,B) but a significant decrease in blood glucose levels in the ITT (effect of Raptor ablation [F_(1, 27)_ = 7.186, *p* = 0.0124]; interaction [F_(5, 135)_ = 3.848, *p* = 0.0027]) and in the area under the curve (AUC) of the ITT, compared to control mice (Figure 3C,D). Female AgRP^ΔRaptor^ mice displayed a slight improvement in glucose tolerance (interaction [F_(5, 145)_ = 4.668, *p* = 0.0006]; Figure 3E,F), although no difference in the ITT was observed between control and AgRP^ΔRaptor^ female mice (Figure 3G,H). Thus, AgRP^ΔRaptor^ mice do not exhibit abnormalities in body growth or energy balance; however, ablation of mTORC1 signaling in AgRP neurons slightly improves glucose homeostasis.

### 3.3. mTORC1 Signaling in AgRP Neurons Does Not Regulate Body Weight during Food Restriction and Refeeding

Previous studies indicate the critical role of AgRP neurons in energy conservation [2], particularly during situations of food restriction [10,23,26,36,37]. To investigate whether mTORC1 signaling in AgRP neurons is required to induce major neuroendocrine adaptations during conditions of negative energy balance, control and AgRP^ΔRaptor^ mice were subjected to 5 days of food restriction (40% of their basal food intake), followed by 5 days of ad libitum refeeding. No differences in food intake were observed between control and AgRP^ΔRaptor^ mice before, during, and after food restriction, either in males (Figure 4A) or females (Figure 4F). Food restriction caused a significant reduction in body weight, fat mass, lean body mass, and glycemia in males (Figure 4B–E) and females (Figure 4G–J). No differences between control and AgRP^ΔRaptor^ mice were observed, except that AgRP^ΔRaptor^ female mice lost less lean body mass compared to control females (interaction [F_(10, 220)_ = 2.563, *p* = 0.006]; Figure 4I). A significant interaction was also observed in the glycemia of female mice (F _(10, 220)_ = 2.876, *p* = 0.0022]; Figure 4J). During the refeeding period, robust hyperphagia was observed in all experimental groups without differences between control and AgRP^ΔRaptor^ mice (Figure 4A,F). Furthermore, the refeeding period was sufficient to recover the reductions in body weight, fat mass, lean body mass, and glycemia previously caused by food deprivation. These changes were similarly observed in control and AgRP^ΔRaptor^ mice, either in males or females. Thus, mTORC1 signaling in AgRP neurons does not modulate the effects of food restriction and refeeding on body weight, body composition, and blood glucose levels. However, the loss of lean mass was attenuated in food deprived AgRP^ΔRaptor^ female mice.

### 3.4. Absence of mTORC1 Signaling in AgRP Neurons Partially Blunts the Reduction in Energy Expenditure Caused by Food Restriction

Food restriction activates AgRP neurons [4,24,25] and produces energy-saving adaptations [10,24,26,31,32]. Consequently, food-deprived animals frequently show reduced VO_2_, a reliable indicator of energy expenditure. AgRP^ΔRaptor^ mice exhibited no differences in VO_2_, RER, and ambulatory activity in the fed state compared to control animals, either in males (Figure 5A–C) or females (Figure 5G–I). Despite the absence of a difference in basal VO_2_, a significant effect of the mutation was observed in the absolute VO_2_ of male AgRP^ΔRaptor^ mice compared to control mice (F_(1, 11)_ = 8.724, *p* = 0.0131), indicating that male AgRP^ΔRaptor^ mice maintain a higher VO_2_ during the food restriction and refeeding periods (Figure 5A). No other differences were observed in the RER or ambulatory activity and in the changes in the VO_2_, RER, and ambulatory activity between male control and AgRP^ΔRaptor^ mice (Figure 5B–F). In the females, we observed a significant interaction in the absolute VO_2_ (F_(10, 110)_ = 2.751, *p* = 0.0046; Figure 5G) and relative VO_2_ (F_(9, 99)_ = 2.704, *p* = 0.0074; Figure 5J), comparing control and AgRP^ΔRaptor^ mice. Particularly in the relative VO_2_, female AgRP^ΔRaptor^ mice presented a blunted reduction in energy expenditure during food restriction compared to control females (Figure 5G). Like the males, no significant differences were observed in the RER (Figure 5H,K) and ambulatory activity (Figure 5I,L) between control and AgRP^ΔRaptor^ female mice.

### 3.5. Neuroendocrine Responses to Food Restriction Are Mildly Attenuated in AgRP^ΔRaptor^ Mice

In the following experiments, the neuroendocrine responses to food restriction were evaluated. As expected [10,24,26,31,32], food restriction increased *Agrp* mRNA expression in the hypothalamus, whereas *Pomc* mRNA levels decreased in food-deprived mice. Nonetheless, no differences between control and AgRP^ΔRaptor^ mice were observed in these responses, either in males or females (Figure 6A–D). Next, the hormonal profile was determined. T4 levels decreased in food-deprived mice, regardless of phenotype or sex (Figure 7A,C). Serum corticosterone levels increased during food restriction (Figure 7B,D). However, although this increase occurred similarly in control and AgRP^ΔRaptor^ male mice (Figure 7B), disruption of mTORC1 signaling in AgRP neurons blunted the increase in serum corticosterone levels of AgRP^ΔRaptor^ female mice (Figure 7D). Thus, neuroendocrine responses to food restriction are mildly attenuated in AgRP^ΔRaptor^ female mice.

### 3.6. AgRP^ΔRaptor^ Mice Show Increased Hyperphagia during Refeeding after an Acute Fasting

In addition to chronic food restriction, acute fasting has been widely used to investigate the role of hypothalamic neurons in regulating food intake [4,14,24,25,31,38,39]. Thus, we determined whether mTORC1 signaling in AgRP neurons is necessary to induce the activation of hypothalamic neurons after an acute fasting (16 h) or following 2 h of refeeding (Figure 8). As previously shown, fasted mice exhibited a prominent Fos expression in the ventromedial ARH (Figure 9A). In contrast, short-term refeeding is sufficient to increase Fos expression in the lateral ARH (Figure 9B). However, these changes were similarly observed in control and AgRP^ΔRaptor^ mice. No differences between fasted and refed mice were observed in the number of Fos-positive neurons in the paraventricular nucleus of the hypothalamus (Figure 9C), lateral hypothalamic area (Figure 9D), and dorsomedial nucleus of the hypothalamus (Figure 9E). Furthermore, disruption of mTORC1 signaling in AgRP neurons did not cause changes in Fos expression in these hypothalamic areas. Refeeding-induced hyperphagia was also evaluated in control and AgRP^ΔRaptor^ mice (Figure 9F). Noteworthy, AgRP^ΔRaptor^ mice showed increased refeeding-induced food intake after an acute fasting (Figure 9F).

## 4. Discussion

In the current study, we investigated the phenotype of mice carrying inactivation of the mTORC1 pathway exclusively in AgRP-expressing cells. Considering the role of mTORC1 signaling in nutrient sensing [11,14] and the importance of AgRP neurons in triggering metabolic adaptations in situations of negative energy balance [10,23,26,36,37], we hypothesized that AgRP^ΔRaptor^ mice may not exhibit a complete metabolic response to food restriction. In accordance with a previous study [22], ad libitum fed AgRP^ΔRaptor^ mice exhibited no major alterations in energy homeostasis. However, AgRP^ΔRaptor^ mice showed a slight improvement in glucose homeostasis. In addition, ablation of mTORC1 signaling affected the changes in energy expenditure during food restriction. Furthermore, female AgRP^ΔRaptor^ mice showed reduced lean body mass loss and a blunted increase in serum corticosterone levels during food restriction.

To demonstrate the efficacy of our cell-specific deletion, we employed a short-term refeeding period as a known activator of the mTORC1 signaling pathway [14,15,22]. Refeeding induced a robust increase in the number of cells expressing pS6 in the ARH of control mice compared to fasted animals. In contrast, the absence of Raptor in AgRP neurons prevented the refeeding-induced increase in pS6-positive cells in the ARH of AgRP^ΔRaptor^ mice. A limitation of our validation is that we were not able to colocalize pS6-positive cells with AgRP neurons in the AgRP^ΔRaptor^ mice. It is well known that using antibodies to label AgRP allows the identification of axons, whereas the staining of cell bodies is poor [5,24]. However, using an AgRP-reporter mouse subjected to refeeding, we observed that approximately 90% of all pS6-positive cells in the ARH were AgRP neurons, demonstrating that the absence of an increase in pS6 cells in the ARH of AgRP^ΔRaptor^ mice can be fully explained by the deletion induced in AgRP cells.

Despite the critical role of hypothalamic mTORC1 signaling in regulating energy homeostasis [14,15,16,17,18,19,20,21,22,23], raptor ablation in AgRP neurons produced mild metabolic effects in ad libitum-fed mice. This phenotype was similar to that observed in a previous study [22] and probably reflects compensatory mechanisms during development that allow AgRP neurons to maintain their physiological functions despite the absence of essential proteins that control their function. This robust plasticity can be exemplified by the fact that early in life, genetic ablation of the leptin receptor gene in AgRP neurons causes only mild consequences for energy homeostasis [40]. In contrast, AgRP-specific leptin receptor deletion in adult animals leads to massive obesity, reproducing the phenotype of *db/db* mice [41]. Another study showed that neonatal ablation of AgRP neurons using diphtheria toxin causes minor effects on energy homeostasis, while the same ablation in adult mice leads to starvation [42]. Thus, using the Cre-LoxP system to induce early-in-life genetic manipulation allows compensatory mechanisms during development, possibly masking the physiological importance of specific proteins.

Although hypothalamic mTORC1 signaling is more related to regulating food intake and energy homeostasis, previous studies have demonstrated that proteins of mTORC1, such as S6K, regulate glucose homeostasis via ARH neurons [15,43]. Overexpression of the p70-S6K1 isoform in the hypothalamus increases fasting glucose levels and hepatic gluconeogenesis in male mice [43]. We observed a slight improvement in the glucose tolerance of female AgRP^ΔRaptor^ mice and the insulin sensitivity of male AgRP^ΔRaptor^ mice. Since AgRP^ΔRaptor^ mice did not exhibit differences in body weight and adiposity compared to control animals, the effects observed in glucose homeostasis were not secondary to changes in body composition.

Chronic food restriction reduces energy expenditure in humans and mice by promoting energy-saving adaptations such as suppression of the thyroid axis, reproduction, and thermogenesis [10,24,26,31,32,44,45]. While male AgRP^ΔRaptor^ mice maintained a higher energy expenditure during the food restriction and refeeding periods than control animals, female AgRP^ΔRaptor^ mice presented a blunted reduction in energy expenditure during food restriction. However, these effects were mild and were not enough to significantly impact the body weight loss of food-deprived mice. Notably, another study showed that mTORC1 signaling in AgRP neurons regulates adaptive energy expenditure by modulating BAT thermogenesis [23]. Thus, our findings and other studies indicate that adaptive energy expenditure is regulated by mTORC1 signaling in AgRP neurons.

We also observed a tendency of male AgRP^ΔRaptor^ mice to present a higher RER during food restriction and refeeding. Chemogenetic activation of ARH^AgRP^ neurons increases carbohydrate utilization while decreasing fat utilization, reflecting an increased RER [37]. Thus, it is possible that mTORC1 disruption may lead to an increased activation of ARH^AgRP^ neurons, explaining the substrate shift to use more carbohydrates as an energy source instead of using fat. We also observed that AgRP^ΔRaptor^ mice showed increased hyperphagia when they regained access to food after a fasting period. In accordance with our findings, inhibition of mTORC1 signaling in the hypothalamus increases food intake by preventing the anorexigenic effect of leptin [14]. Fasting-induced hyperphagia is associated with the activity of ARH^AgRP^ neurons [4,25,31,32]. So, our findings suggest that disruption of mTORC1 signaling may increase the activity of ARH^AgRP^ neurons, although our approaches to indirectly assess the activation of these neurons (e.g., Fos expression or gene expression of *Agrp* or *Npy* transcripts) were not sensitive enough to detect statistically significant differences between the groups.

Hypothalamic *Agrp* expression is upregulated in food-deprived animals, whereas *Pomc* expression is suppressed [4,10,24,26,31,32]. These results were reproduced in our study in both males and females and were not affected by mTORC1 disruption in AgRP neurons. Serum T4 levels are decreased during food restriction, indicating a well-established inhibition of the thyroid axis [26]. In contrast, serum corticosterone levels increase in food-deprived animals as a stress response and to maintain endogenous glucose production in the absence of enough food. While food restriction-induced suppression of serum T4 levels was not significantly altered in mice carrying Raptor ablation in AgRP neurons, female AgRP^ΔRaptor^ mice showed lower serum corticosterone levels during food restriction than food-deprived control mice. Previous studies have shown that AgRP neurons can regulate corticosterone secretion, corticotrophin-releasing hormone (CRH) expression, or neuron activity [10,26,38,39]. For example, chemogenetic inhibition of AgRP neurons reduces CRH content in the paraventricular nucleus of the hypothalamus [39]. Another study has shown that AgRP neuron activation increases circulating corticosterone levels [38]. Thus, our study indicates that mTORC1 signaling in AgRP neurons regulates food-restriction-induced glucocorticoid secretion in female mice. The mechanisms behind this sex difference are unknown; however, robust evidence suggests that ARH neurons are sexually dimorphic [27]. Furthermore, the circulating levels of hormones that regulate food intake, the sensitivity to these hormones, the responsiveness to a high-fat diet, and many other metabolic aspects present significant sex differences [24,28,29,30]. ARH^AgRP/NPY^ neurons do not express estrogen receptor α [46]. However, changes in estrogen levels cause profound effects on *Npy* and *Agrp* gene expression and the fasting-induced activation of ARH^AgRP/NPY^ neurons [46]. We also observed that only female AgRP^ΔRaptor^ mice showed a reduced lean mass loss during food restriction compared to control females. Glucocorticoids enhance protein degradation in skeletal muscle, favoring muscle mass loss in catabolic conditions [47,48]. Thus, it is possible that the blunted increase in corticosterone secretion in female AgRP^ΔRaptor^ mice during food restriction may have spared their loss of lean body mass. However, future studies are necessary to test this possibility. Female AgRP^ΔRaptor^ mice also exhibited a more substantial decline in glycemia at the beginning of the food restriction. The blunted increase in corticosterone secretion during food restriction may also explain the lower glycemia of female AgRP^ΔRaptor^ mice on the first day of food restriction compared to control females.

## 5. Conclusions

Our findings confirm our initial hypothesis that mTORC1 signaling in AgRP neurons regulates, to some extent, the metabolic and neuroendocrine responses to food restriction. However, the phenotype of AgRP^ΔRaptor^ mice is mild, and the differences observed were sex dependent. Nonetheless, we provide evidence that early-in-life disruption of mTORC1 signaling in AgRP neurons is sufficient to produce significant metabolic alterations in mice.

## Figures and Tables

**Figure 1 cells-12-02442-f001:**
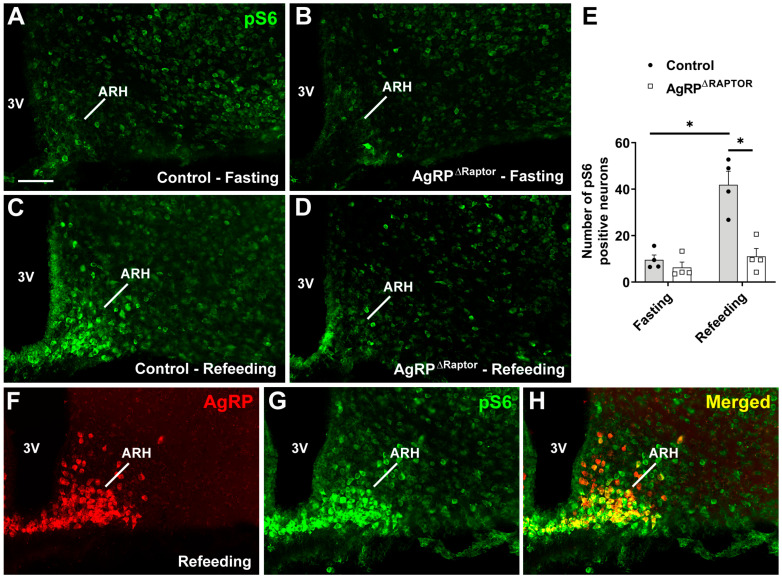
Generation of mice carrying ablation of the mTORC1 signaling in ARH^AgRP^ neurons. (**A**–**D**) Epifluorescence photomicrographs showing pS6 immunoreactive cells in the ARH of fasted control mice (**A**), fasted AgRP^ΔRaptor^ mice (**B**), refed control mice (**C**), and refed AgRP^ΔRaptor^ mice (**D**). (**E**) Quantification of the number of pS6-positive cells in the ARH (*n* = 4/group). (**F**–**H**) Epifluorescence photomicrographs showing the colocalization between AgRP (indirectly visualized by a reporter protein) and pS6 in the ARH of an AgRP-reporter mouse subjected to refeeding. Abbreviation: 3V, third ventricle. Scale bar = 100 µm. *, *p* < 0.05 (two-way ANOVA, followed by the Newman–Keuls multiple comparisons test).

**Figure 2 cells-12-02442-f002:**
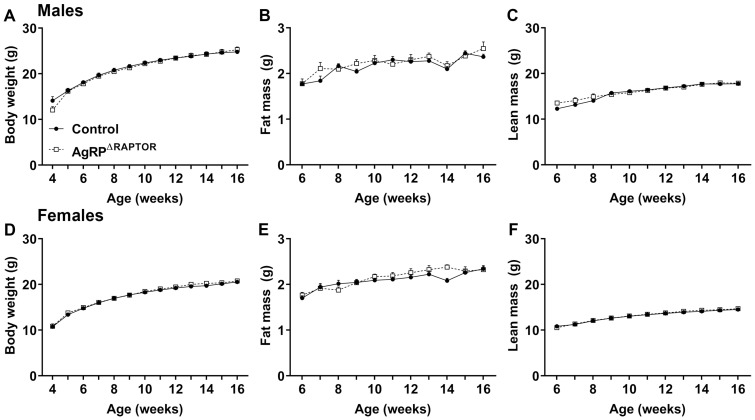
AgRP^ΔRaptor^ mice exhibit no changes in body weight. (**A**–**C**) Changes over time in body weight, fat mass, and lean body mass in control (*n* = 15) and AgRP^ΔRaptor^ (*n* = 12) male mice. (**D**–**F**) Changes over time in body weight, fat mass, and lean body mass in control (*n* = 25) and AgRP^ΔRaptor^ (*n* = 15) female mice. Data were analyzed by repeated measures two-way ANOVA.

**Figure 3 cells-12-02442-f003:**
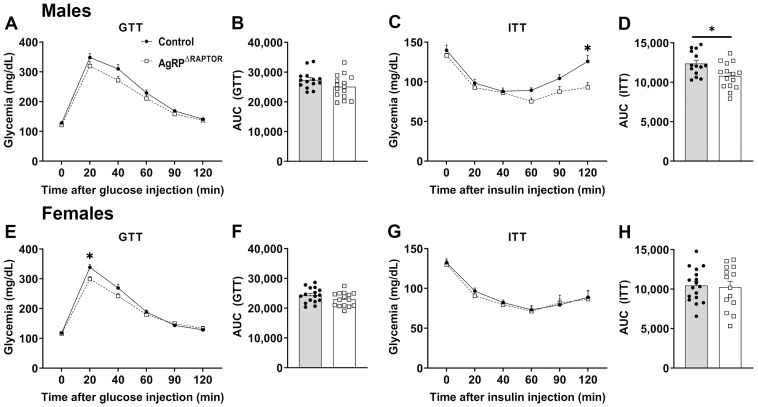
AgRP^ΔRaptor^ mice show a slight improvement in glucose homeostasis. (**A**–**D**) Blood glucose levels and the area under the curve (AUC) of the glucose tolerance test (GTT) and insulin tolerance test (ITT) in control (*n* = 14) and AgRP^ΔRaptor^ (*n* = 15) male mice. (**E**–**H**) Blood glucose levels and the AUC of the GTT and ITT in control (*n* = 16) and AgRP^ΔRaptor^ (*n* = 14) female mice. Changes in blood glucose levels were analyzed by repeated measures two-way ANOVA (and Sidak’s multiple comparisons test). An unpaired, two-tailed Student’s *t*-test was used to analyze the AUC. *, *p* < 0.05, significantly different compared to control mice.

**Figure 4 cells-12-02442-f004:**
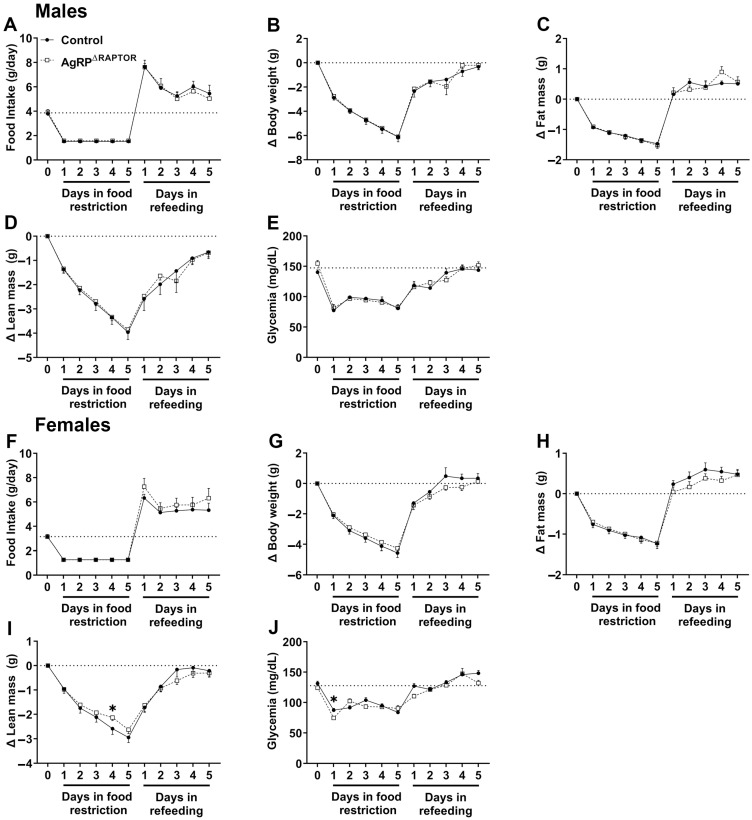
mTORC1 signaling in AgRP neurons does not regulate body weight during food restriction and refeeding. (**A**–**E**) Changes in food intake, body weight, fat mass, lean body mass, and glycemia during food restriction and refeeding in control (*n* = 8) and AgRP^ΔRaptor^ (*n* = 10) male mice. (**F**–**J**) Changes in food intake, body weight, fat mass, lean body mass, and glycemia during food restriction and refeeding in control (*n* = 12) and AgRP^ΔRaptor^ (*n* = 12) female mice. Data were analyzed by repeated measures, two-way ANOVA, and Sidak’s multiple comparisons test. *, *p* < 0.05, significantly different compared to control mice.

**Figure 5 cells-12-02442-f005:**
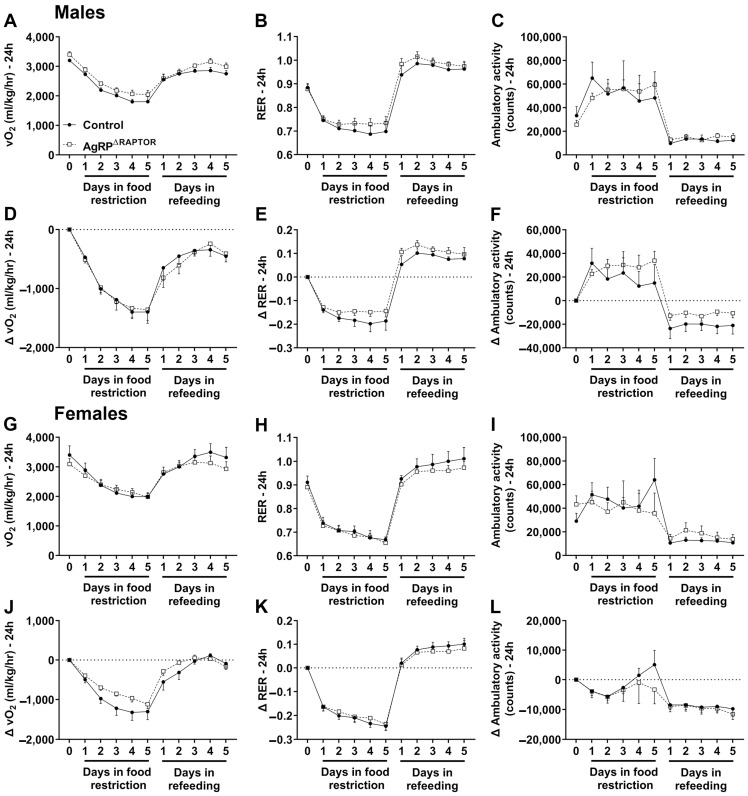
The absence of mTORC1 signaling in AgRP neurons partially blunts the reduction in energy expenditure caused by food restriction. (**A**–**F**) Changes in VO_2_, respiratory exchange ratio (RER), and ambulatory activity, either in absolute values or relative to baseline, during food restriction and refeeding in control (*n* = 6) and AgRP^ΔRaptor^ (*n* = 7) male mice. (**G**–**L**) Changes in VO_2_, RER, and ambulatory activity, either in absolute values or relative to baseline, during food restriction and refeeding in control (*n* = 7) and AgRP^ΔRaptor^ (*n* = 7) female mice. Data were analyzed by repeated measures, two-way ANOVA, and Sidak’s multiple comparisons test.

**Figure 6 cells-12-02442-f006:**
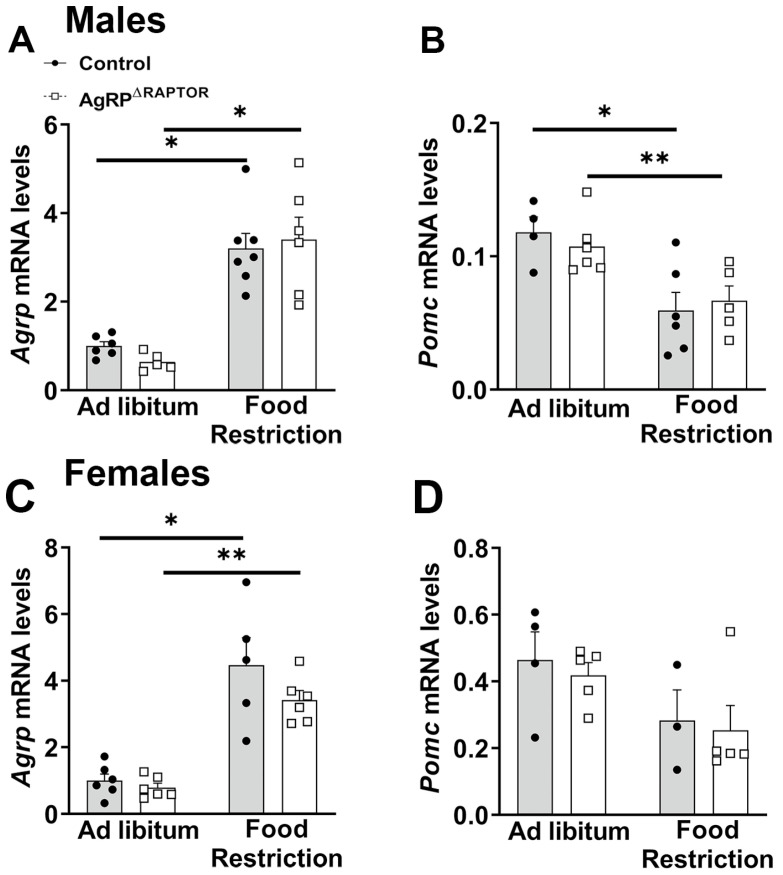
Hypothalamic mRNA levels in ad libitum fed and food-restricted mice. (**A**–**D**) Relative mRNA expression in control and AgRP^ΔRaptor^ male and female mice, either in ad libitum fed conditions or after 2 days of food restriction (*n* = 4–7/group). Data were analyzed by a two-way ANOVA and a Newman–Keuls multiple comparisons test. *, *p* < 0.05; **, *p* < 0.01 significantly different compared to control mice.

**Figure 7 cells-12-02442-f007:**
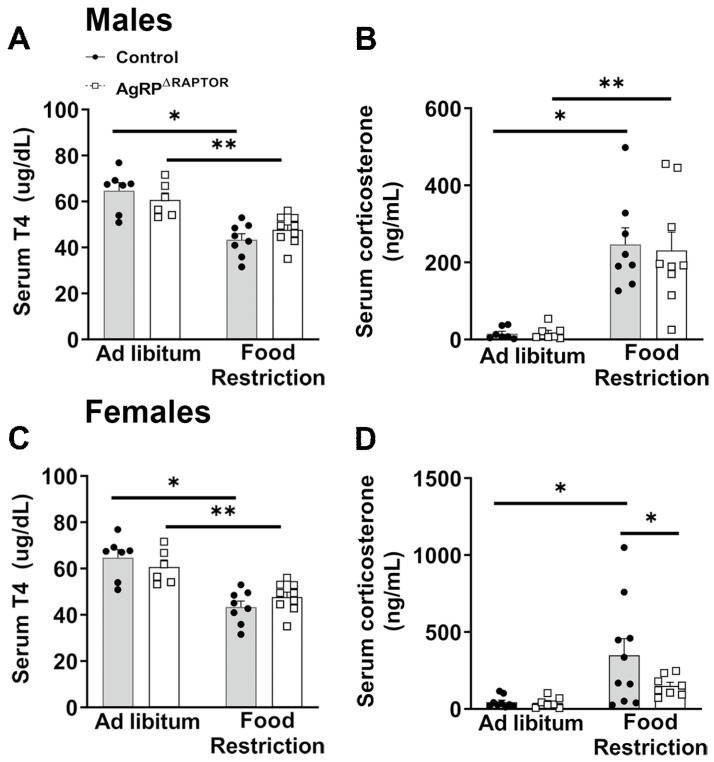
Hormonal profile in ad libitum fed and food-restricted mice. (**A**–**D**) Serum T4 and corticosterone levels in control and AgRP^ΔRaptor^ male and female mice, either in ad libitum fed conditions or after 2 days of food restriction (*n* = 7–10/group), were analyzed by a two-way ANOVA and a Newman–Keuls multiple comparisons test. *, *p* < 0.05; **, *p* < 0.01 significantly different compared to control mice.

**Figure 8 cells-12-02442-f008:**
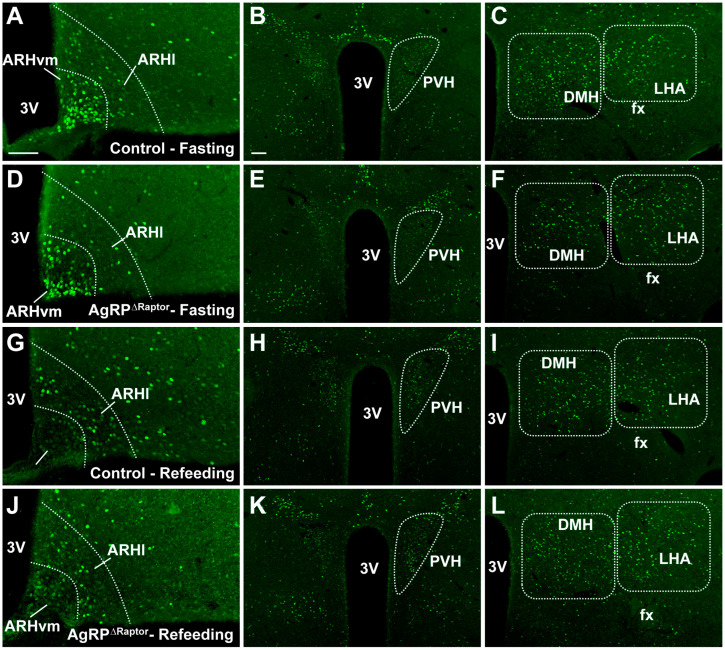
Expression of Fos protein in the hypothalamus of fasted and refed mice. (**A**–**L**) Epifluorescence photomicrographs showing Fos immunoreactive cells in different hypothalamic nuclei of fasted control mice (**A**–**C**), fasted AgRP^ΔRaptor^ mice (**D**–**F**), refed control mice (**G**–**I**), and refed AgRP^ΔRaptor^ mice (**J**–**L**). Abbreviations: 3V, third ventricle; ARHl, lateral arcuate nucleus; ARHvm, ventromedial arcuate nucleus; DMH, dorsomedial nucleus of the hypothalamus; fx, fornix; LHA, lateral hypothalamic area; PVH, paraventricular nucleus of the hypothalamus. Scale bars = 100 µm.

**Figure 9 cells-12-02442-f009:**
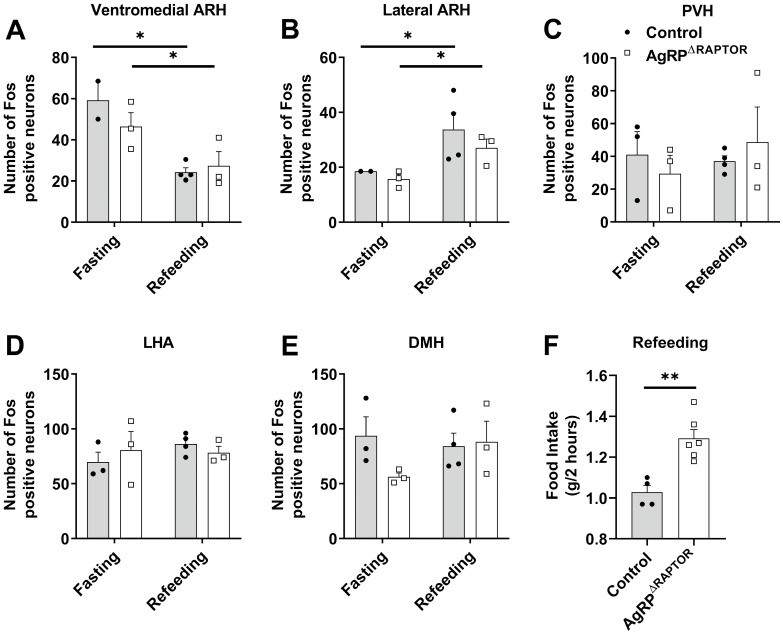
Effects of fasting and refeeding on the activation of hypothalamic neurons and food intake. (**A**–**E**) Quantification of the number of Fos-positive neurons in the hypothalamus (*n* = 3–4/group). (**F**) Food intake during 2 h of refeeding after an overnight fasting (16 h). Data were analyzed by a two-way ANOVA, a Newman–Keuls multiple comparisons test, or an unpaired two-tailed Student’s *t*-test (food intake). *, *p* < 0.05; **, *p* < 0.01 significantly different compared to control mice.

## Data Availability

The data that supports the findings of this study are available from the corresponding author upon reasonable request.

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
