# Peer review of "mTORC1 Signaling in AgRP Neurons Is Not Required to Induce Major Neuroendocrine Adaptations to Food Restriction"

_cells, 2023, doi:10.3390/cells12202442_

Round 1

Reviewer 1 Report

This extremely-well written manuscript by de Souza et al. describes an impressive and detailed investigation on the role of mTORC1 in AgRP neurons in the control of metabolic homeostasis. The authors selectively delete mTORC1 from AgRP neurons using Cre/lox technology in mice and perform various experiments to determine changes in feeding behavior, glucose metabolism, energy expenditure, and circulating hormones. While some of the findings relevant to the basic metabolic phenotype confirms previously published work, findings related to food restriction are novel and of great interest.

There are two major concerns that, if addressed, would largely strengthen the conclusions to be drawn from the study.

1.     The authors nicely demonstrate that AgRP-ΔRaptor mice show a profoundly diminished expression of pS6-expressing cells in the brain region where AgRP neurons are located. While this confirms the efficiency (and selectivity) of the genetic approach, it remains unclear whether and how pS6 expression relates to the activity regulation of AgRP neurons. Specifically, does lack of mTORC in AgRP neurons impair their ability to increase neuronal activity upon food restriction? Demonstrating this would be critical as it would directly link the mTOR pathway to energy-state-dependent activity regulation of AgRP neurons – a key mechanism for behavioral and metabolic adaptations during starvation. To test this, the author could consider determining Fos expression in AgRP neurons in fed and fasted/food-restricted mice. Given that AgRP neurons control activity in various brain regions through their broad projections, examining Fos expression in these downstream areas in control versus AgRP-ΔRaptor mice would further strengthen the manuscript.

2.     The authors use AgRP-ires-Cre mice for the deletion of mTORC in AgRP neurons. While this is a cell-type specific approach, it comes with a caveat: mTORC is lacking from birth on, and already during development. Given that ablation of AgRP neurons in adult mice largely impairs food intake, but fails to affect basic feeding patterns when the ablation is performed in neonatal mice (PMID: 16254186), it could be that compensatory mechanisms occur during development when AgRP neurons are not present. Further studies provide evidence that Cre/lox-dependent deletion of key genes from AgRP neurons (for example the Leptin receptors) also fails to impact metabolic homeostasis. However, deletion of the same genes from adult mice causes massive alterations (PMID: 29670283 and 32699414). Thus, it is possible that mTORC deletion from AgRP neurons in adult animals would cause very different (much stronger) phenotypes. There are several genetic approaches the authors could use to address this critical issue (CRISPR-based editing, inducible Cre-expressing mice, Flp- or Dre-dependent strategies). 

Author Response

This extremely-well written manuscript by de Souza et al. describes an impressive and detailed investigation on the role of mTORC1 in AgRP neurons in the control of metabolic homeostasis. The authors selectively delete mTORC1 from AgRP neurons using Cre/lox technology in mice and perform various experiments to determine changes in feeding behavior, glucose metabolism, energy expenditure, and circulating hormones. While some of the findings relevant to the basic metabolic phenotype confirms previously published work, findings related to food restriction are novel and of great interest.

There are two major concerns that, if addressed, would largely strengthen the conclusions to be drawn from the study.

Response: We thank the reviewer for the time spent evaluating our study and for the constructive comments that improved the manuscript.

  1. The authors nicely demonstrate that AgRP-ΔRaptor mice show a profoundly diminished expression of pS6-expressing cells in the brain region where AgRP neurons are located. While this confirms the efficiency (and selectivity) of the genetic approach, it remains unclear whether and how pS6 expression relates to the activity regulation of AgRP neurons. Specifically, does lack of mTORC in AgRP neurons impair their ability to increase neuronal activity upon food restriction? Demonstrating this would be critical as it would directly link the mTOR pathway to energy-state-dependent activity regulation of AgRP neurons – a key mechanism for behavioral and metabolic adaptations during starvation. To test this, the author could consider determining Fos expression in AgRP neurons in fed and fasted/food-restricted mice. Given that AgRP neurons control activity in various brain regions through their broad projections, examining Fos expression in these downstream areas in control versus AgRP-ΔRaptor mice would further strengthen the manuscript.

Response: We would like to thank the reviewer for this suggestion. In the revised manuscript, we included new data (Figure 8 and Figure 9) showing the expression of Fos, a marker of neuronal activation, in the arcuate nucleus (either in the ventromedial or lateral parts), paraventricular nucleus of the hypothalamus, lateral hypothalamic area, and dorsomedial nucleus of the hypothalamus (projecting areas of AgRP neurons) of fasted and refed mice. Fasted mice exhibited higher Fos expression in the ventromedial arcuate nucleus, whereas refeeding increased the expression of Fos in the lateral arcuate nucleus. However, these differences were similarly observed in control and AgRPΔRaptor mice. No significant differences between the groups were also observed in the number of Fos-positive cells in the paraventricular nucleus of the hypothalamus, lateral hypothalamic area, and dorsomedial nucleus. In the revised manuscript, we also analyzed refeeding-induced hyperphagia in control and AgRPΔRaptor mice. Noteworthy, AgRPΔRaptor mice showed increased refeeding-induced food intake after an acute fast. The new results were included and discussed in the revised manuscript.

  1. The authors use AgRP-ires-Cre mice for the deletion of mTORC in AgRP neurons. While this is a cell-type specific approach, it comes with a caveat: mTORC is lacking from birth on, and already during development. Given that ablation of AgRP neurons in adult mice largely impairs food intake, but fails to affect basic feeding patterns when the ablation is performed in neonatal mice (PMID: 16254186), it could be that compensatory mechanisms occur during development when AgRP neurons are not present. Further studies provide evidence that Cre/lox-dependent deletion of key genes from AgRP neurons (for example the Leptin receptors) also fails to impact metabolic homeostasis. However, deletion of the same genes from adult mice causes massive alterations (PMID: 29670283 and 32699414). Thus, it is possible that mTORC deletion from AgRP neurons in adult animals would cause very different (much stronger) phenotypes. There are several genetic approaches the authors could use to address this critical issue (CRISPR-based editing, inducible Cre-expressing mice, Flp- or Dre-dependent strategies).

Response: We appreciate the reviewer´s suggestion of using different approaches that avoid compensatory mechanisms during development to investigate the importance of mTORC signaling in AgRP neurons. However, it would be necessary to redo the entire study using, for example, virus-induced genetic manipulation to target the Raptor gene in the hypothalamus of adult mice. We are currently unable to repeat all the experiments using this methodology. Anyhow, the possibility that compensatory mechanisms during development may have masked the importance of mTORC signaling in AgRP neurons for the regulation of energy homeostasis was detailed discussed in the manuscript (third paragraph of the Discussion section):

“Despite the critical role of hypothalamic mTORC1 signaling in regulating energy homeostasis [14-23], Raptor ablation in AgRP neurons produced mild metabolic effects in ad libitum-fed mice. This phenotype was similar to that observed in a previous study [22] and probably reflects compensatory mechanisms during development that allow AgRP neurons to maintain their physiological functions despite the absence of essential proteins that control their function. This robust plasticity can be exemplified by the fact that early in life genetic ablation of the leptin receptor gene in AgRP neurons causes only mild consequences on energy homeostasis [36]. In contrast, AgRP-specific leptin receptor deletion in adult animals leads to massive obesity reproducing the phenotype of db/db mice [37]. Another study showed that neonatal ablation of AgRP neurons using diphtheria toxin causes minor effects on energy homeostasis, while the same ablation in adult mice leads to starvation [38]. Thus, using the Cre-LoxP system to induce early-in-life genetic manipulation allows compensatory mechanisms during development, possibly masking the physiological importance of specific proteins.”

Reviewer 2 Report

The manuscript by De Souza et al investigates the role of raptor in AGRP-mediated phenotypes in both male and female mice. The rationale behind the paper is well-explained and the experiments itself are well designed. Additionally, a lot of studies do not look at gender differences, and I applaud the group for reporting results from both male and female mice. However the most obvious drawback of the paper is the lack of a striking phenotype with AGRP-raptor mice. So this makes it important to potentially run some additional experiments to add more weight to the paper. Anyway, below are my comments-

1.     You show very nice immunofluorescence images refed conditions. Under the methods section, you mention that for IF these mice have been fasted overnight, and then refed for 2 hours. But as I gather, for food intake studies that you showed, the design was a food restriction for 5 days followed by refeeding. Why not show IF images for food restriction followed by refeeding since the food intake studies were done under these conditions?

2.     In the same vein, since your effects of food restriction followed by refeeding showed no dramatic changes on food intake, did you run any additional experiments with mice being fasted overnight or less, followed by refeeding? Or even food intake studies ranging from 1-24 hours under dark or light cycle without fasting? 

3.     I am all for publishing negative data as well as positive data. So I like that the authors do include negative data in this manuscript. However, I am not sure why the authors looked at body composition when body weights were not different. Is it not expected that fat or lean mass is not changed when body weights are similar?

4.     Is the slight improvement in GTT due to increased insulin release after a glucose bolus? Have you looked at insulin levels or leptin or any additional hormones?

5.     Since you see higher RER, have you looked at fat oxidation graphs? Although you do not see a change in adiposity, you observe a slightly higher RER in males. Higher RER (or a value closer to 1), indicates a shift in substrate utilization from fat to carbs. So in essence you have lesser reliance on fats being used as a primary fuel source which translates into a higher fat mass. There is evidence on AGRP regulating substrate utilization which you cite in your manuscript, but please take a look at the paper again- Regulation of substrate utilization and adiposity by Agrp neurons, nat comm, 2019. 

6.     You see some very interesting gender-specific differences in your model. But you hardly focus on this in your manuscript. There is only a few lines at the very end of your discussion. Sexual dimorphism or ARC neurons is a very important topic, and your results further highlight this. So please write more about this in your introduction as well. Please refer to a recent review published in Cells- Neurochemical Basis of Inter-Organ Crosstalk in Health and Obesity: Focus on the Hypothalamus and the Brainstem, Cells, 2023. 

7.     While female mice do not become obese, have you run any additional experiments with male mice on a high fat diet? Several times the phenotypes become more apparent when mice become obese as opposed to when they are lean?

Author Response

The manuscript by De Souza et al investigates the role of raptor in AGRP-mediated phenotypes in both male and female mice. The rationale behind the paper is well-explained and the experiments itself are well designed. Additionally, a lot of studies do not look at gender differences, and I applaud the group for reporting results from both male and female mice. However the most obvious drawback of the paper is the lack of a striking phenotype with AGRP-raptor mice. So this makes it important to potentially run some additional experiments to add more weight to the paper. Anyway, below are my comments-

Response: We thank the reviewer for the time spent evaluating our study and for the constructive comments that improved the manuscript.

  1. You show very nice immunofluorescence images refed conditions. Under the methods section, you mention that for IF these mice have been fasted overnight, and then refed for 2 hours. But as I gather, for food intake studies that you showed, the design was a food restriction for 5 days followed by refeeding. Why not show IF images for food restriction followed by refeeding since the food intake studies were done under these conditions?

Response: To validate the mTORC1 disruption in AgRP neurons, we used acute fasting, followed by refeeding, to assess pS6 expression. Our option to use this protocol was based on previous studies that showed that fast/refeeding acutely modulates the activity of mTORC1 pathway (Cota et al., Science 312:927-930, 2006; Albert et al., Biochem Biophys Res Commun 464:480-486, 2015; Burke et al., eLife 6:e22848, 2017). On the other hand, our research group has more experience in assessing the role of AgRP neurons on the metabolic adaptations to food deprivation using chronic calorie restriction (Furigo et al., Nat Commun 10:662, 2019; Furigo et al., J Mol Endocrinol 64:13-27, 2020; Pedroso et al., J Nutr Biochem 84:108457, 2020). That is the reason for using 5 days of calorie restriction to determine whether AgRPΔRaptor mice show a different response compared to control animals. In the revised manuscript, we included new data (Figure 8 and Figure 9) showing the expression of Fos, a marker of neuronal activation, in the arcuate nucleus (either in the ventromedial or lateral parts), paraventricular nucleus of the hypothalamus, lateral hypothalamic area, and dorsomedial nucleus of the hypothalamus (projecting areas of AgRP neurons) of fasted and refed mice. Fasted mice exhibited higher Fos expression in the ventromedial arcuate nucleus, whereas refeeding increased the expression of Fos in the lateral arcuate nucleus. However, these differences were similarly observed in control and AgRPΔRaptor mice. No significant differences between the groups were also observed in the number of Fos-positive cells in the paraventricular nucleus of the hypothalamus, lateral hypothalamic area, and dorsomedial nucleus.

  1. In the same vein, since your effects of food restriction followed by refeeding showed no dramatic changes on food intake, did you run any additional experiments with mice being fasted overnight or less, followed by refeeding? Or even food intake studies ranging from 1-24 hours under dark or light cycle without fasting?

Response: We appreciate the reviewer´s suggestion. We performed additional experiments to determine food intake during 2 hours of refeeding in mice previously subjected to fasting (16 hours). Noteworthy, AgRPΔRaptor mice showed increased refeeding-induced food intake after an acute fast. These new results (Figure 9F) were included and discussed in the revised manuscript.

  1. I am all for publishing negative data as well as positive data. So I like that the authors do include negative data in this manuscript. However, I am not sure why the authors looked at body composition when body weights were not different. Is it not expected that fat or lean mass is not changed when body weights are similar?

Response: Since we had the possibility of analyzing the body composition longitudinally, we believed that the inclusion of such information (fat mass and lean body mass) would add valuable information to the manuscript, even though no changes in body weight were observed. Actually, there are situations in which changes in both components of body composition, however inversely, may result in no changes in body weight. Thus, if only body weight is analyzed, these metabolic alterations cannot be seen. For example, we have shown that prepubertal leptin-deficient (ob/ob) mice do exhibit differences in body weight compared to wild-type mice. However, the absence of changes in body weight is masked by decreased lean mass that compensates for the increased body adiposity exhibited by ob/ob mice (Teixeira et al., J Endocrinol 249:239-251, 2021).

  1. Is the slight improvement in GTT due to increased insulin release after a glucose bolus? Have you looked at insulin levels or leptin or any additional hormones?

Response: Unfortunately, we were unable to determine insulin or leptin serum levels in our study.

  1. Since you see higher RER, have you looked at fat oxidation graphs? Although you do not see a change in adiposity, you observe a slightly higher RER in males. Higher RER (or a value closer to 1), indicates a shift in substrate utilization from fat to carbs. So in essence you have lesser reliance on fats being used as a primary fuel source which translates into a higher fat mass. There is evidence on AGRP regulating substrate utilization which you cite in your manuscript, but please take a look at the paper again- Regulation of substrate utilization and adiposity by Agrp neurons, nat comm, 2019.

Response: We would like to thank the reviewer for the recommendation to discuss in more details the aforementioned study in light of our findings. Since the RER results only showed tendencies and not statistically significant results, we were careful in not overdiscussing them. However, considering your suggestion, we added a phrase in the revised manuscript discussing this tendency and taking into consideration the findings published by Cavalcanti-de-Albuquerque et al. (2019) (pages 14-15):

“We also observed a tendency of male AgRPΔRaptor mice to present a higher RER during food restriction and refeeding. Chemogenetic activation of ARHAgRP neurons increases carbohydrate utilization (while decreasing fat utilization), reflecting an increased RER [37]. Thus, it is possible that mTORC1 disruption may lead to an increased activation of ARHAgRP neurons, explaining the substrate shift to use more carbohydrates as an energy source instead of using fat. We also observed that AgRPΔRaptor mice showed increased hyperphagia when they regained access to food after a fasting period. In accordance with our findings, inhibition of mTORC1 signaling in the hypothalamus increases food intake by preventing the anorexigenic effect of leptin [14]. The fasting-induced hyperphagia is associated with the activity of ARHAgRP neurons [4,25,31,32]. So, our findings suggest that disruption of mTORC1 signaling may increase the activity of ARHAgRP neurons, although our approaches to indirectly assess the activation of these neurons (e.g., Fos expression or gene expression of Agrp or Npy transcripts) were not sensitive enough to detect statistically significant differences between the groups.”

  1. You see some very interesting gender-specific differences in your model. But you hardly focus on this in your manuscript. There is only a few lines at the very end of your discussion. Sexual dimorphism or ARC neurons is a very important topic, and your results further highlight this. So please write more about this in your introduction as well. Please refer to a recent review published in Cells- Neurochemical Basis of Inter-Organ Crosstalk in Health and Obesity: Focus on the Hypothalamus and the Brainstem, Cells, 2023.

Response: We agree with the reviewer that we only briefly discussed the importance of sex differences in our manuscript, despite having observed many differences between males and females. Thus, we added phrases in the Introduction and Discussion sections, expanding the discussion about the importance of sex differences in studying metabolism, and cited the Haspula & Cui (2023) paper in the revised manuscript.

  1. While female mice do not become obese, have you run any additional experiments with male mice on a high fat diet? Several times the phenotypes become more apparent when mice become obese as opposed to when they are lean?

Response: We agree with the reviewer that the observation of phenotypes sometimes requires metabolic challenges, such as exposition to high fat diet. However, we decided to focus our current study in investigating situations of negative energy balance, which are known to induce the activation of AgRP neurons. Furthermore, the importance of mTORC1 signaling in AgRP neurons for the metabolic responses to HFD were already evaluated in a previous study (Albert et al., Biochem Biophys Res Commun 464:480-486, 2015).

Round 2

Reviewer 2 Report

The authors have answered all my questions/concerns. No further comments